# Digital Quality Resources Resulting from Standardized Program for Rubric Training in Medical Residents

**DOI:** 10.3390/healthcare10112209

**Published:** 2022-11-03

**Authors:** María Valeria Jiménez Báez, María Erika Gutiérrez De la Cruz, María Margarita Chávez Hernández, Luis Roberto Martínez Castro, Francisco Javier Alcocer Nuñez

**Affiliations:** 1Coordination of Planning and Institutional Link, Mexican Institute of Social Security in Quintana Roo, Cancun 77533, Mexico; 2Social Service Intern Physician, Mexican Institute of Social Security in Quintana Roo, Cancun 77533, Mexico

**Keywords:** scoring, rubrics, health personnel, program evaluation, distance education, residency education, specialty

## Abstract

Introduction: In the health care area, tuition is an essential part to provide the instrument that proves the graduates have acquired the necessary skills in their specialties. Objective: Evaluation of the improvement in quality of resources for residents after standardized digital training program with rubrics. Methods: Prospective observational study of first year medical residents in seven medical specialties in four different training centers. Five dimensions were considered to scale the quality of medical resident research: Validation of rubric in investigation methodology topics for each block in b-learning mode; initial and ending evaluation; colloquium investigation rubric; results of final investigation; satisfaction survey of 360 degrees. The instruments were validated using the delphi method with a minimum agreement of 0.8. We considered global values greater than 80 points as good quality. Results: 85 medical residents participated and obtained a final average of 80.62 (±9.59), and the satisfaction of the course was qualified as excellent/good in 82.5%. A positive relation was observed between the scope of the evaluation and the level of satisfaction. Mean quality score for the course was good. There is no relationship between the research experience of the students and the final average r = 0.123 (*p* = 0.291). Conclusions: The implementation of research seminars in b-learning mode results in improving the education program for health residents after a training program with a rubric system and their acquiring research skills, and, as a consequence, the final product also showed better quality, even when the student did not have any experience in a scientific publication.

## 1. Introduction

The medical residency is the set of academic, care and research activities that the resident doctor must carry out within the medical units during the time that corresponds to the operational programs [1]. Resident doctors are part of the base of the National Health System in Mexico because they offer specialized services for complex health problems [2].

In the training period, the resident needs to comply with the activities and regulations stipulated by the Hospital and they also, like students of the University, should complete the calendar of all the activities mandated by the studies plan. [2,3]. The research module in the plan is found transversally during the training years of medical specialists.

Research is one of the fundamental aspects of the training for doctors, who are intended to develop analytical skills and be able to produce an original investigation project during their residency and, as a consequence, to contribute to improve the scientific knowledge in their specialist field. [4].

All research projects should be published in a scientific indexed journal according to Herrera-Añazco et al. In their study, 1062 residents registered with the National Council of Resident Physicians (CONAREME) were evaluated and it was observed that 11.1% published some article during residency [5]. Although most training centers for medical specialties teach subjects related to the development of skills and competencies in research, even the process by which the impact, training and time invested by the resident are assessed is deficient [6].

Teaching in health care is a pillar of training and should provide the appropriate instruments to guarantee graduates with the required skills, including the generation and broadcasting of research in their area [7], and within it, the evaluation is essential in the teaching–earning process, which still remains as a challenge [8].

Miller’s pyramid [9] considers four steps for the acquisition of professional skills. The first step refers to “knowing” what can be examined through written evaluation tools; the second, “knowing how to apply”, can be evaluated with a written test; in the third step, “show how it would be done”, the evaluations are performed with simulators whose objective is to try to reproduce real-life situations in standardized conditions that allow observers to analyze and evaluate the action in this context, including the Objective Structured Clinical Examination (OSCE) [10], simulations by computer and standardized patient simulations. The fourth step is “doing”, as is the case with medical action in practice. For this purpose, evaluation methods need instruments such as: review of medical records, observation of practice by other colleagues or through videos, patient or relative satisfaction surveys, portfolios, and quality of therapeutic prescriptions, among others [7]. In health research, the fourth step of the Miller pyramid should be objectified during training. One way proposed by various health educators has been the inclusion of rubrics [11].

The subject of health research has always represented an arid field for students who found it boring and difficult to conduct, which is a reason for why motivation is a relevant element of the outcome. The consequence of lack of interest is a poor project at the end of the year in the residency. The purpose of this study is to evaluate the improvement in health research after a rubric training program. 

## 2. Materials and Methods

This study is a prospective observational study carried out in the state of Quintana Roo, México in public health hospitals, which involved four different medical units with residencies and seven medical specialties during the first year of the research module.

### 2.1. Participants

We considered as a universe the residents studying a specialty in the medical care public hospitals of Quintana Roo and selected randomly the academic year of the specialty. The first year was selected and included all the first-degree residents from four residence halls, enrolled during the 2021–2022 academic cycle at the Mexican Institute of Social Security (IMSS) in the state of Quintana Roo. It was not necessary to calculate the sample size since 100% of the first-year medical residency fellows were included.

### 2.2. Sources of Eligibility Criteria for the Selection of Participants

As inclusion criteria to participate in this study, they were considered to be a regular student in specialization courses at the IMSS OOAD Quintana Roo and to be enrolled in the first-year research course on the CPEI Moodle platform. Non-inclusion criteria: those students who did not correspond to the grade, or who were external to the unit, even if they were from the same grade. Elimination criteria: Those students who did not carry out academic activities in a proportion greater than 20%.

### 2.3. Variables

The following variables were established and defined:

Results: the quality of the research of the medical residents in research courses with the learning method was evaluated on 5 dimensions (Figure 1).

Theoretical course: It was delivered through the Moodle^®^ platform and had a duration of 8 weeks with a weighting of 50% of the final grade of the course, a total of 16 activities distributed with their corresponding evaluation rubric. The rubrics were subjected to pair review with 100% agreement. Each activity had a defined time for its preparation and delivery with a minimum duration of 2 days and a maximum of 2 weeks (depending on the complexity of the activity).Colloquium: It consisted of a face-to-face presentation of the research project that aimed to demonstrate the theoretical knowledge acquired in front of experts (minimum of 2) who evaluated through a standardized rubric with a weighting of 20% in the final gradeEnd product: In the last stage of the seminar, the fellows delivered their complete research protocol, which was sent to the local research committee for approval and subsequent application of the measurement instruments. It was evaluated with standardized rubrics for qualitative evaluation and to ensure the acquisition of knowledge, and validated by expert peer researchers, with a weighting of 30%.Post-Seminar Cognitive Assessment. Finally, we applied a post-seminar test of 26 questions with the relevant topics that the fellows studied throughout the research seminar and thus evaluated the acquisition of skills and abilities that each fellow obtained.Satisfaction survey: 22 questions divided into 4 sections were applied: 1. Satisfaction related to the student; 2. Satisfaction with the development of the course; 3. Satisfaction with the teachers of the course; and 4. Satisfaction related to the development of the program. This survey was designed and validated by experts in the field using the Delphi method with a 3/3 concordance. This method is characterized by using a structure to obtain the degree of consensus or agreement of specialists on a given problem. The process of this method is based on the following characteristics: selection criteria: health professionals with master’s degrees and/or doctorates in science and previous publications on the subject; number of experts: no number was recommended. However, three experts were included in this study; iterative process in rounds: in this study, four rounds of revision were carried out to reach a consensus; criteria to consider for the completion of the process: the consensus was estimated with mean and percentage [12]. Therefore, the survey allowed us to evaluate the level of quality and satisfaction that the fellows perceived about the different aspects that made up the research course.

The research trajectory of the fellows was considered as an intervening variable in the study: it was determined by the experience of the student as referred to in reply to a direct question, considering as trajectory having completed a thesis, research project or original articles in scientific journals before the specialty.

### 2.4. Evaluation Rubrics

Evaluation instruments previously reviewed by authorities and adapted to the corresponding seminar were designed for each topic and specific skill included the course program, standardized and validated by expert researchers using the Delphi panel method. Post-seminar evaluation: instruments were used at the end of the seminar that allowed evaluating the knowledge acquired. Sociodemographic variables such as age, medical specialty, and satisfaction with the course were considered.

### 2.5. Design and Validation of Academic Content of the Module

The academic topic was integrated by modules according to the academic program of each specialty, which was designed to develop a research protocol at the end of the course. For the validation, there was a working group of experts made up of a teacher of more than 25 years of the research module, a doctor in research education of the national level I system, and a thesis student from the medical career as a validating peer for research seminar development. Electronic searches were performed in the Educational Resources Information Center (ERIC) with a search strategy based on key concepts of interest for the objectives: evaluation rubrics, medical specialties, educational program, virtual learning, and pandemic education. A restriction of publication dates was imposed with a range of 5 years to the current year (2017–2021). The contents were added based on the training programs contemplated by the educational institutions, after reviewing five educational institutions with medical specialty programs, standardizing the content of the “research methodology seminar”. For the creation and subsequent standardization of the evaluation rubrics, three rounds were carried out with feedback and corrections by each expert up to a 3/3 concordance.

### 2.6. Statistical Analysis

We integrated the data to a base of the SPSS system: “Statistical Package for the Social Sciences” (SPPS version 21). Descriptive statistics were applied: measures of central tendency including mean, median, standard deviation, proportions or percentages, which were determined with the Chi-square or Fisher test for qualitative variables, and the Student’s t-statistic was used as a hypothesis test for the bivariate analysis and ANOVA of one factor for different groups, considering the grouping variables satisfied a minimum significance of 0.05. A Spearman’s correlation test was estimated for the final product and the publishing experience. The results are expressed in the tables.

### 2.7. Ethical Aspects

The procedures adhere to the ethical standards, the Regulations of the General Health Law on Research for Health, and the Declaration of Helsinki and its amendments. In the present study, the institutional provisions and those of the General Health Law 18 and the Mexican Official Standard are adhered to. Therefore, it was approved by the Research Ethics Committee and the Local Health Research Committee.

## 3. Results

A total of 85 first-year medical residents (*n* = 85) enrolled in the research seminar with 100% participation in the period from 14 August to 1 October 2021 were registered.

The mean age was 31.34 years (SD ± 3.96); for men it was 38 years (SD ± 3.13) and women, 31.51 years (SD ± 3.833). When classifying the fellows by specialty, the highest frequency was for Family Medicine with 35 (41.2%) physicians (Table 1).

The mark of the theoretical course on the Moodle platform was 78.75 (±10.45), with the specialty of family medicine being the highest with 83.31 (±10.51) and anesthesiology the lowest performing with 71.71 (±9.50). For the colloquium category, an average score of 75.25 (±27.66) can be observed, with Internal Medicine being the most relevant with 92.48 (±4.48), and the lowest performance being Emergency Physicians with 27.78 (±35.95). Regarding the category of the quality of the final product, the mean was 87.29 (±7.96) and better performance was observed for the specialty of pediatrics 92.50 (±9.57) and the lowest performance was for medical emergencies—surgical with 80.00. The final degree, made up of the items of the theoretical course, colloquium, and final product, had a global grade of 80.61 (±9.59) with a minimum of 49.16 and a maximum of 95.16; the one with the best performance was pediatrics with 85.82 (±4.37). The ratings of the post-seminar cognitive evaluation by specialty are as follows: The highest average was Family Medicine 77.58 (±8.21), Internal Medicine 72.73 (±7.78) and Emergency Physicians with 66.45 (±11.88) was the lowest average. Pearson’s correlation was 0.709. (*p* = 0.001) (Table 2).

A division of groups was made with respect to the level of care of the specialty under study and a difference was observed in the means of the qualification of each one of the dimensions evaluated (Table 3).

Regarding experience in research methodology, the following results were found: 8 medical residents with publication, 38 medical residents with formal education in research methodology, 2 medical residents who took an introductory course in research methodology and 37 medical residents without methodological background. A Spearman–Brown correlation was performed between this intervening variable and the rating of the final product with a value of 0.123; *p* = 0.26. In the same way, the comparison of medians between the intervening variable and the qualification of the final production was carried out, with no statistical difference being observed between the group of medical residents with a history of scientific publication and the group without publication; *p* = 0.92. (Figure 2)

When the evaluation of the quality of the final work was performed through the rubric, a lower performance was observed in the section referring to “Material and methods” and in “Bibliographical references in Vancouver Format”; in both cases, they were not structured according to the guidelines granted on the Moodle^®^ platform.

A satisfaction survey was applied with a Cronbach’s alpha consistency of 0.742 in 74 (87.1%) fellows. In the first section, 47 (63.5%) refer to satisfaction related to the student as “good”. In the second section, the global satisfaction of the course was referred to as good in 62.2%. A positive relationship was observed between the scope of evaluation obtained in the subject and the level of satisfaction evaluated by the student (Table 4).

## 4. Discussion

Rubrics are guides with a specific criteria list. Although they are used in the educational field for a wide variety of tasks, they are most often used to assess the work of students and teachers. For the teacher, they are useful tools since they allow planning and analyzing the expectations of the class, course or tasks. Within the functions of the rubrics, the following stand out: (1) The determination of the degree or level of mastery that a student has reached in a certain task, measuring it as: excellent, good or insufficient; (2) Measurement: A rubric allows measurement by assigning a score to each expectation, goal, or criterion listed in the rubric; (3) They explain the expectations of students, parents, other teachers, and others about homework expectations; (4) Self-assessments to record individualized progress [13].

In the time of the SARS-CoV-2 pandemic, the face-to-face teaching modality was limited. Nearly 200 countries activated maximum alerts so that educational centers were completely or partially closed; as a consequence, the academic training of nearly 1600 million people at different educational levels was affected [13]. In the case of higher education and postgraduate courses, most universities used digital platforms as a complement to the development of teaching [13,14]. When we evaluated residents about virtual education and simulation tools during the COVID-19 pandemic, 74.1% considered them adequate; however, 75.9% agreed that the decrease in practice activities had an impact in the hospital management of patients, particularly in invasive or surgical procedures [15].

The Research Seminar is part of the Single Program of Medical Specialties (PUEM); in a transversal way, it is located in the curricular map throughout the duration of the program of each specialty, regardless of whether it is medical or surgical [4]. In this research, it was proposed to prove that the exercise of education through a virtual classroom with the implementation of a course with b-learning modality and a standardized evaluation system with rubrics favorably affects the quality of research products and improves competencies.

For the training of the medical resident in research, it is necessary to contextualize that the traditional development of the career included an exclusively face-to-face approach to the curricular contents as a consequence of the development of information technologies and the growing need to achieve research products as an essential requirement. However, to obtain the title of specialist, we are faced with the need to find a route that allows us to develop a research project that is of quality, and in response this project was designed.

There are interventions related to improving the learning of medical residents in the search for strategies that improve the acquisition of skills; an example of this is a study by the Faculty of Medicine of the National Autonomous University of Mexico that described the experience of a b-learning program of the research seminar for medical residents given from 2010 to 2016, referring to a pass rate of 92.4% with an overall rating of 89.0 (±1.09). The program also included the implementation of didactic guides for the elaboration of the final product, continuous training for tutors in the management of the Moodle platform, updating of resources for medical residents and a tutoring model with a pedagogical reference, Advisor-Tutor-Resident (ATR), to improve online tutoring aimed at the teaching process. [16] Although there are already references that tried to demonstrate the positive influence of the quality of the research derived from distance courses, the global response of health institutions and educational institutions forced both systems to implement distance programs in a timely manner during the emergency. Thus, it is important to evaluate the quality of the products that emanate from this type of distance education.

Research itself is part of one of the three pillars of health professional training and in terms of the area of medicine and medical specialties, it is present within the undergraduate and postgraduate study plans; however, only 11.1% have scientific publication during residency even though 100% of specialists trained in the area of medicine have research projects [5]. Therefore, when we analyzed the quality of these research projects as a complex construct formed from the generation process during the research seminars, we were able to approach the understanding of this phenomenon; there is a long way to go so that research becomes an ability, dexterity and competence applicable in a useful way by the medicine specialist. It is necessary that the health system, together with the teaching staff and the managers of the training programs of medical specialties, standardize the elements aimed at improving the quality of educational programs and obtaining research products.

Comparing the results obtained in our study with those referred to, we found that the overall rating was 80.61 (±3.96), slightly lower than that obtained by Ponce et al. [16]. During the development of the course, a guide-manual was developed for the development of the research protocol. However, a weakness observed was that during the specialty programs, the student ideally requires a clinical tutor accompanied by a methodological tutor; despite the fact that there are clinical teaching staff, few have training for the research area. There are proposals that this weakness is secondary to the lack of interpersonal motivation for research and the conception that it is a tedious process; others refuse to take courses of this nature for various reasons, such as loss of interest and lack of time, to name some.

It has been mentioned that the teacher’s satisfaction improves the development of the course, as in the study carried out by González Martínez [17]; likewise, it was observed that obtaining better results in the course increases the level of satisfaction (*p* < 0.05). It is important to point out that in our study, it was observed that four out of ten scholarship holders report little satisfaction with the development of the course, considering in this dimension that the questions were directed to the fulfillment of the objectives of the course, the organization and the planning of the contents of the course, the workload of each topic during the course and the active participation of the students during the course. It is important to highlight that these results allow us to have an approximation to how the student perceives the course; however, as teachers, it will be necessary to consider that satisfaction is sometimes influenced by the student’s behavior, intrinsic motivation and the scope obtained on their own products, and that at the time of being submitted to an interview or questionnaire, the student responds exclusively to their own consideration. So, it is important that greater objectivity be incorporated in satisfaction studies as much as possible.

The use of rubrics was fundamental for the evaluation on time of each activity in this research course. Their application was from this year, although certain rubrics were already applied in isolated modules. The evaluation of the final product was not concreted and shows that there were many disagreements between tutors and interns. Uriah et al. [18] points out that evaluation rubrics offer a positive impact on the learning process, due to the standardization of criteria, and Morales-López et al. [19], mentions that they are one way to transmit quality criteria for tasks and thus improve the quality of activities. In our study, the standardized rubrics were a resource with great educational potential. They allowed us to intertwine the relationship between the learning, teaching and evaluation process, while clarifying which phases of the elaboration of the final product were less useful, as referred to by Cockett, when identifying the usefulness of the rubrics for evaluation to determine their critical thinking and case-solving ability through positive feedback [20].

It stands out as a strength of this study to have carried out the prior standardization of teachers, professors and tutors with rubrics, allowing the actors of the educational process to clarify the abilities, knowledge and skills that are intended to be developed during the course.

Experience in a scientific publication prior to the medical residency is not a guarantee of obtaining quality research products in the research seminar of the medical residency courses; therefore, it is necessary for research professors of medical residency courses to design comprehensive content focused on motivating and developing research skills in medical residents in their professional work and not making an inference between experience and a higher qualification.

There will be students who, upon entering their training as medical specialists, have experience in editing; however, as teachers, the challenge is to avoid assuming that these students already have adequate competence to conduct research.

The general average of the final product rating was 87.29 (±7.96), so the quality of the research protocols was considered good. There is a direct relationship between the degree of satisfaction and the final score, since the higher the score, the greater the satisfaction, as is the case for family medicine and internal medicine. Flores K. et al. [21] mention that it is necessary to pay attention to the multiple criteria that make up the quality of a b-learning course and its activities.

It is necessary that educational planners, when training human resources for health, specifically medical specialists, consider the incorporation into the curricular design of subjects that in a transversal way contribute to the development of investigative competence in the future specialist, including the skills of critical reading, literature evaluation and decision making.

The generation of knowledge requires sustained work on the part of the student but also accompaniment during this process by the system that forms it, in which directors, teachers and evaluators participate with a holistic approach, and the systems, instruments, competencies and individual and collective values are not seen as separate entities, but are combined into one overarching concept [22]. In this study, we clearly observe that there is a positive impact of having structured evaluations; however, by including the evaluation of satisfaction as an essential component within this subject, it allows teachers to identify areas of greater need according to that of students, and with this to improve these different areas.

The acquisition of investigative skills is complex, but it is necessary for trainers to consider it as an essential requirement and promote it in the training processes of medical specialists, which naturally benefit patients.

In the age of digitization, as a natural tool in training, it is necessary for medical specialists, teachers and educational managers to incorporate it into their educational processes since current generations have this technological competence inherent in their environment.

## 5. Conclusions

Online education became the recourse of postgraduate education institutions, aware that the educational needs and expectations of students and society in general cannot be adequately met in a face-to-face modality due to the actual epidemiological context.

Although the relevance of this modality continues to be studied, in this study, we concluded that the delivery of the research seminar in a b-learning modality as a response to the need for training in medical residents in the area of health education showed a higher qualification and greater satisfaction. Moreover, we determined that comprehensive evaluation through the use of standardized rubrics allowed delimiting the deficiencies and strengths of timely feedback, influencing the process of acquiring skills and the quality of the final product.

There are still inefficiencies in the program. However, the combination of students’ and tutors’ perspectives helped to provide feedback and improve our research seminar to offer even higher quality health education and research.

## Figures and Tables

**Figure 1 healthcare-10-02209-f001:**
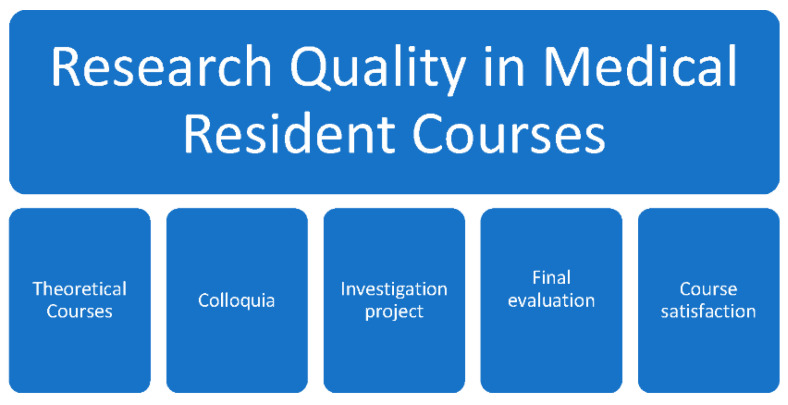
Dimensions: research quality.

**Figure 2 healthcare-10-02209-f002:**
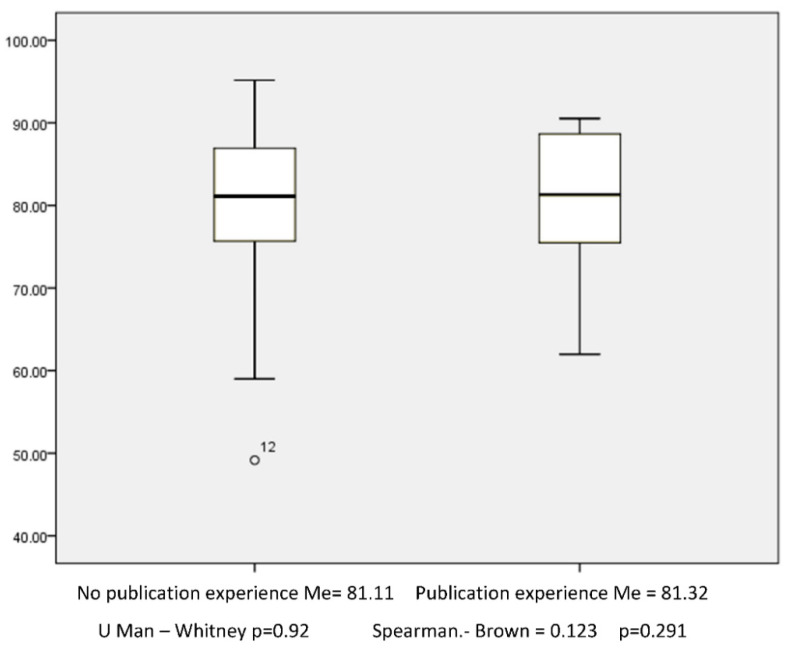
Publicized experience in the resident like a variable of the final evaluation.

**Table 1 healthcare-10-02209-t001:** Sex distribution and medical specialty.

	Sex		Total
	Male	Female	
Anesthesiology	8	3	11
	21.10%	6.40%	12.90%
Gynecology and Obstetrics	1	4	5
	2.60%	8.50%	5.90%
Radiology	4	5	9
	10.50%	10.60%	10.60%
Family Medicine	16	19	35
	42.10%	40.40%	41.20%
Internal Medicine	5	6	11
	13.20%	12.80%	12.90%
Pediatric	1	3	4
	2.60%	6.40%	4.70%
Emergency Physicians	3	7	10
	7.90%	14.90%	11.80%
Total	38	47	85
	100.00%	100.00%	100.00%
	44.70%	55.30%	100.00%

Fisher test *p* > 0.05.

**Table 2 healthcare-10-02209-t002:** Score from different products.

	Moodle µ (±) IC. LI-LS	Colloquiums µ (±) IC. LI-LS	Final Product µ (±) IC. LI-LS	Final Score µ (±) IC. LI-LS	Post-Seminar Cognitive Assessment µ (±) IC. LI-LS
Anesthesiology (*n* = 11)	71.70 (±9.5)	71.84 (±24.38)	84.55 (±5.22)	75.58 (±9.95)	70.63 (±7.74)
	IC. 65.32–78.08	IC. 55.46–88.22	IC. 81.04–88.05	IC. 68.90–82.27	IC. 65.42–75.83
Gynecology and Obstetrics (*n* = 5)	78.85 (±6.27)	67.47 (±9.98)	80.00 (±0.00)	76.92 (±2.64)	72.31 (±9.58)
	IC. 71.07–86.64	IC. 55.07–79.86	IC. 80.00–80.00	IC. 73.64–80.20	IC. 60.42–84.20
Radiology (*n* = 9)	75.94 (±9.5)	80.26 (±30.45)	86.11 (±7.82)	79.86 (±6.44)	71.37 (±9.25)
	IC. 67.12–84.76	IC. 56.86–103.66	IC. 80.10–92.12	IC. 74.91–84.80	IC. 64.26–78.47
Family Medicine (*n* = 35)	83.30 (±10.51)	82.60 (±16.66)	89.57 (±8.61)	85.04 (±7.81)	77.58 (±8.21)
	IC. 79.69–86.92	IC. 76.88–88.32	IC. 86.61–92.53	IC. 82.36–87.73	IC. 74.76–80.40
Internal Medicine (*n* = 11)	75.98 (±9.79)	92.48 (±4.48)	91.82 (±6.03)	84.03 (±6.13)	72.73 (±7.78)
	IC. 69.40–82.56	IC. 89.46–95.49	IC. 87.77–95.87	IC. 79.91–88.15	IC. 67.50–77.95
Pediatric (*n* = 4)	80.07 (±10.47)	90.18 (±2.67)	92.50 (±9.57)	85.82 (±4.37)	75.96 (±7.28)
	IC. 63.40–96.73	IC. 85.93–94.42	IC. 77.27–107.73	IC. 78.86–92.76	IC. 64.37–87.55
Emergency Physicians (*n* = 10)	75.61 (±7.40)	27.78 (±35.95)	80.00 (±0.00)	67.36 (±9.67)	66.45 (±11.88)
	IC. 70.31–80.90	IC. 2.06–53.50	IC. 80.00–80.00	IC. 60.44–74.28	IC. 57.95–74.94
Total (*n* = 85)	78.76 (±10.46)	75.26 (±27.67)	87.29 (±7.96)	80.62 (±9.59)	73.70 (±9.27)
	IC. 76.50–81.01	IC. 69.29–81.22	IC. 85.58–89.01	IC. 78.54–82.69	IC. 71.70–75.70

µ: Mean, ±: Standard deviation, IC95%: Confidence Interval, LI: Lower Limit, LS: Upper Limit. The Pearson correlation was 0.709. (*p* = 0.001).

**Table 3 healthcare-10-02209-t003:** Score from different products in two groups.

Course Dimensions	No Family Physician Residents Specialist		Family Physician Residents Specialist		* *p*
	*n* = 50		*n* = 35		
	Mean	SD	Mean	SD	
Moodle	75.57	9.25	83.31	10.51	0.00
Colloquiums	70.11	32.46	82.60	16.66	0.00
Final Product	85.70	7.14	89.57	8.61	0.00
Final score	77.52	9.57	85.04	7.81	0.00
Post-Seminar Cognitive Assessment	70.98	9.07	77.58	8.21	0.00

* *t*-test for Quality Components and Specialty Courses.

**Table 4 healthcare-10-02209-t004:** Satisfaction frequency by dimension and globally.

	Excellent µ (%) IC. LI-LS	Good µ (%) IC. LI-LS	Bad µ (%) IC. LI-LS
Satisfaction Related to the Student	14 (18.9%)	47 (63.5%)	13 (17.6%)
	IC. 10.8–27.0	IC. 54.1–75.4	IC. 8.3–25.4
Satisfaction with the Development of the Course	18 (24.3%)	26 (35.1%)	30 (40.5%)
	IC. 13.7–35.1	IC. 24.3–47.3	IC. 26.4–50.9
Satisfaction with the Teachers of the Course	18 (24.3%)	37 (50.0%)	19 (25.7%)
	IC. 15.1–33.8	IC. 37.0–62.2	IC. 17.6–36.5
Satisfaction Related to the Development of the Program	24 (32.4%)	44 (59.5%)	6 (8.1%)
	IC. 21.6–44.6	IC. 47.3–68.9	IC. 2.9–13.5
Course Satisfaction Level	15 (20.3%)	46 (62.2%)	13 (17.6%)
	IC. 12.2–31.1	IC. 50.2–71.4	IC. 6.1–27.0

µ: Mean, %: Percentage, IC95%: Confidence Interval, LI: Lower Limit, LS: Upper Limit. ANOVA Test for Quality Components and Specialty Courses, *p* = 0.001.

## Data Availability

Not applicable.

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
