# Peer review of "Digital Quality Resources Resulting from Standardized Program for Rubric Training in Medical Residents"

_healthcare, 2022, doi:10.3390/healthcare10112209_

Round 1

Reviewer 1 Report

The present paper taps on a really important aspects of education in medicine: The teaching of research metodology. The study is conducted on a cohort of first years specialty students of a public health services institustion.  Participants number is adequate and variables operazionalization is accurate. Nevertheless the study exhibit critical points that must be solved before publication.

First of all, even if I am not an English mother tonges speaker I am pretty sure that the manuscript need to be linguistically revised. 

In general term, the aim of the paper in somewhat ambiguous. Is it purely descriptive of an experience of Distance Learning making use of evaluation rubrics or it pretend to demonstrate the efficacy of such an approach? If the second is the case, this paper fails its aim.

There is no clear research hypothesis (which is quite surprising given the topic of the e-Learning course). If the authors disambiguate the aim of the study they will easyly able to formulate a clear research hypotesis. The result section reproduce the same uncertainty. Depending on the way the research hypotresis is formulated, some results are central and other ancillary. This section must be re-written.

In particular:

The abstract don’t need to be so detailed: what is the paper about, what have been done and the main results achieved. Drop out the specifications on raters' agreement to the questionnaire emploied, levels of statistical significance, software adopted for the analysis and so on.

Page 2: line 2 delete the character “4”.

Page 4: line 13 I can’t see any t test or ANOVA in the result section.

Later in the same paragraph: there are no histogram of graphs in the version of the paper I’ve received.

Page 4, result section: I see no reason to put in the text and in a table the same information. A short sentence inviting the reader to look at the table would be better.

Page 5, result section: again, leave the table and omit numbers in the text

bottom of page 5: the authors refer to Table 2 as reporting the result of a pearson correlation but in fact the table contains other information

page 6, first paragraph: the authors describe participants’ previouse competence in research methodology. Does it have an impact on evaluation rubric? Please compute the opportune statistical test and describe the results properly.

Page 6: once again the text and the table contains the same information

After reading the results section, the reader remains with an anunswered question: “is this b-learning modality theaching more or less effective than other approaches to the theaching of research methodology in medicine?”

Pag 7, last paragraph: you pretend that there is a direct relationship between course satisfaction and  the final score but no specific analysis is provided in the result section.

Author Response

good evening.
the document is sent with the requested revisions.

Reviewer 2 Report

The study tried to address two components: standardized digital training and an evaluation system with rubrics. However, more details of the standardized evaluation system need to be clearly written. The language for quantitative analyses needs further improvement, please refer to the writing manual for statistics. For example, there are multiple places, either correlation or ANOVA test significance p=0.001, is it exactly p=0.001 or is it p < 0.001 or p< .01 or p < .05.  The notes under the table are not well written for readers to understand, e.g., “ANOVA Test for Quality Components and Specialty Courses p=0.001”, are you saying ANOVA Test for each Quality Component between Specialty groups are significant at p <.001?

The analyses were very descriptive, without any prior context about the training in traditional modality, I am not convinced you have fully answered the research question of “a virtual classroom with the implementation of a course with b-learning modality and a standardized evaluation system with rubrics favorably affects the quality of the research products and improve the investigative skills of the specialist doctor who is being trained.” I am not saying you have to have a comparison group, but some context about the training quality in the past still would help the reader better understand the implications of this digital training program and standardized evaluation system.

I am providing more detailed suggestions below for improvement.

Abstract: I don’t think you need list all the demographics details of the sample

Page 3: you mentioned the intervening variable—research background, but not clearly to me if you have taken account of that variable in your analyses. I know you talked in page 6 that it has no relationship with the final projects, which makes me wondering what about other quality components.

There are places that need to check the grammar, for example, page 2: “Teaching in medicine is very important must offer the appropriate instruments to demonstrate that graduates have acquired the necessary skills [7].” In the abstract, you have multiple future tense “will” which is not appropriate in that the study has been completed. English editing service might be needed.

Page 3 you briefly mentioned the standardized Rubric was validated by experts using Delpha-panel method, you did not provide any citation to it, I would assume this is the first time to be used for research purposes. So more details about the rubric are needed, e.g., how it was developed and validated,  what are the criteria and rating scales etc.

Ethic aspect should be condensed to very brief description of Institutional Review Board (IRB) approval or similar.

You don’t need to repeat the sample size in one sentence, get rid of n=85 in multiple places: “85 first-year medical residents (n=85) …”  

Table 1, you claimed there is no difference for sex according to specialty. But I saw the percentage of male and female were quite different for Anesthesiology compared to other specialty.

Page 5:  the narrative in the results is repetitive of results in the table 2. You should be summarizing the importance findings and may also talked about post hoc test on the specialty differences of the ANOVA test.

Table 3: I am not clear what those numbers (e.g., IC. 10.8-27.0 vs. IC. 54.1-75.4) are measuring, please either explain in the narrative clearly or make clear notes under the table 3. Also,  how does the rating scale look like and how many questions in each section of the satisfaction survey should be added to the methods section about the survey.

Author Response

Response to Reviewer 2 Comments

Point 1: The study tried to address two components: standardized digital training and an evaluation system with rubrics. However, more details of the standardized evaluation system need to be clearly written. The language for quantitative analyses needs further improvement, please refer to the writing manual for statistics. For example, there are multiple places, either correlation or ANOVA test significance p=0.001, is it exactly p=0.001 or is it p < 0.001 or p< .01 or p < .05.  The notes under the table are not well written for readers to understand, e.g., “ANOVA Test for Quality Components and Specialty Courses p=0.001”, are you saying ANOVA Test for each Quality Component between Specialty groups are significant at p <.001?

Response 1: The table where the ANOVA description was found has been eliminated at the suggestion of reviewer 1 and it has been decided to describe the results.

Point 2: The analyses were very descriptive, without any prior context about the training in traditional modality, I am not convinced you have fully answered the research question of “a virtual classroom with the implementation of a course with b-learning modality and a standardized evaluation system with rubrics favorably affects the quality of the research products and improve the investigative skills of the specialist doctor who is being trained.” I am not saying you have to have a comparison group, but some context about the training quality in the past still would help the reader better understand the implications of this digital training program and standardized evaluation system.

Response 2:

The Research Seminar is part of the Single Program of Medical Specialties (PUEM), in a transversal way it is located in the curricular map during the entire duration of the specialty program in question, regardless of whether it is medical or surgical [15]. In this research, it is proposed to prove that the exercise of education through a virtual classroom with the implementation of a course with b-learning modality and a standardized evaluation system with rubrics favorably affects the quality of research products and improves skills. investigative. of the medical specialist in training. It is necessary to put in context that the traditional development of the program included an approach to curricular content exclusively in person as a result of the development of information technologies and the growing need to achieve research products as an essential requirement to obtain the degree of specialist. , we are faced with the need to find a route that allows residents to obtain two objectives, to develop a research project and for this project to be of quality, in response to which this project was designed.

Point 3: Abstract: I don’t think you need list all the demographics details of the sample

Response 3: Demographic data from the sample has been removed.

Point 4: Page 3: you mentioned the intervening variable—research background, but not clearly to me if you have taken account of that variable in your analyses. I know you talked in page 6 that it has no relationship with the final projects, which makes me wondering what about other quality components.

Response 4:

It is noteworthy that 10.8% of the scholarship holders report having some type of publication before the seminar, 51.4% report only having studied subjects related to re-search methodology, 2.7% report having completed workshops or introductory courses on the methodology, and 35.1% report not having any type of background on research methodology, without statistical significance (p>0.5) that is related to the quality of the final projects.  It can be seen that there is no relationship between the research back-ground of the students or in any dimension of the quality of the projects generated during the course.

Point 5: There are places that need to check the grammar, for example, page 2: “Teaching in medicine is very important must offer the appropriate instruments to demonstrate that graduates have acquired the necessary skills [7].” In the abstract, you have multiple future tense “will” which is not appropriate in that the study has been completed. English editing service might be needed.

Response 5: The suggested grammar has been modified and the possible errors that may be found are analyzed in detail.

Point 6: Page 3 you briefly mentioned the standardized Rubric was validated by experts using Delphi-panel method, you did not provide any citation to it, I would assume this is the first time to be used for research purposes. So more details about the rubric are needed, e.g., how it was developed and validated,  what are the criteria and rating scales etc.

Response 6: Added more information regarding the Delphi method.

Point 7: Ethic aspect should be condensed to very brief description of Institutional Review Board (IRB) approval or similar.

Response 7: The description of the ethical aspects has been modified as requested.

Point 8: You don’t need to repeat the sample size in one sentence, get rid of n=85 in multiple places: “85 first-year medical residents (n=85) …” 

Response 8: Sample number n=85 has been removed

Point 9: Table 1, you claimed there is no difference for sex according to specialty. But I saw the percentage of male and female were quite different for Anesthesiology compared to other specialty.

Response 9: Said observation has been modified, emphasizing anesthesiology residents

Point 10: Page 5:  the narrative in the results is repetitive of results in the table 2. You should be summarizing the importance findings and may also talked about post hoc test on the specialty differences of the ANOVA test.

Response 10: Table 2 has been eliminated at the suggestion of reviewer 1 and the description of the results remains, highlighting the most relevant.

Point 11: Table 3: I am not clear what those numbers (e.g., IC. 10.8-27.0 vs. IC. 54.1-75.4) are measuring, please either explain in the narrative clearly or make clear notes under the table 3. Also,  how does the rating scale look like and how many questions in each section of the satisfaction survey should be added to the methods section about the survey.

Response 11: The decision was made to eliminate table 3 and leave the description of the results.

Reviewer 3 Report

Healthcare 2021, 9.

Quality of research in residents of medical specialties after a standardized digital training program with rubrics.

The article is clear and well written. Easy to read and understand, perhaps due to its linearity. It is observed that authors lacks critical thinking that would have made the article more interesting. Likewise, it can be affirmed that there is interesting information that is not provided and that means that the complexity of the processes it analyzes is not appreciated. For example:

The authors explain that “They used the Miller pyramid, hierarchized with four steps that range from cognition to behavior in professional practice, the first step refers to "knowing", the second to "knowing how to apply", the third to "show how it will be done" and the fourth is to "do" as is the case of medical performance in practice. The rubrics are a guide that lists the specific criteria,” We would have liked to know more about the rubrics and how they have applied the steps of the Miller pyramid.

A second example is the lack of more information about the students' context that has produced this statement: “The subject of health research has always represented an arid field for the student population, so its development is more related to the motivation of that acquisition” We would like some argument as to why this arid field arises. Don't received students enough capacitation and motivation in the grade?

Subsequently, they tell us about five dimensions of the quality research evaluation: the theoretical course, the colloquiums, the final product, the post-seminar evaluation, and the satisfaction survey. And also the design of the Evaluation Rubrics as an Evaluation Instrument composed of a rubric for each topic and specific skill. Presenting some example of the rubrics would have clarified the process, as well as some information on the Post-Seminar Assessment Tool.

In the Results section, a table is shown in relation to the medical specialties, when the following table, of the results of the satisfaction survey, is performed globally. They also do not justify why it is done in this way since no argument is derived regarding the results by specialty, we see no reason to present the tables differently.

In the presentation of the averages obtained from the population studied, the reason for the low average number of students with publications is not commented: “It is noteworthy that 10.8% of the scholarship holders report having some type of publication before the seminar”.

The authors express that “When evaluating the quality of the final project through the rubric, a lower performance was observed in the section referring to "Material and methods". And in the Discussion authors argument “the student ideally requires a clinical tutor accompanied by a methodological tutor, although there is clinical teaching staff, few have the training for the research area.” Given the importance of mastering research methodologies for young researchers at the beginning, and this being a weak point, this fact is not deepened, as it should.

Additionally, we find scant mention of the following result, “30 (40.5%) scholarship holders refer to their satisfaction with the development of the course as "bad"". But in the discussion, it is pointed out that “In turn, the satisfaction survey allowed us to visualize the highlights that need to be fed back to improve the research seminar: the format and practicality of the contents, the clarity, and diversity of the activities, the structure of the course, and the interactivity between teachers and students”. No results have been shown in these aspects! The discussion and the conclusions are excessively optimistic: The results of the satisfaction surveys do not contain reasons why the students have expressed that their satisfaction is excellent, good or bad, therefore we do not understand where they have obtained the information to affirm that “However, there are still deficiencies in the program, which thanks to a combination of the perspective of the student and the teacher, allows us to provide feedback and improve our research seminar to provide even higher quality in health education and research”.  We do not doubt the following two statements of the authors:

“Standardized rubrics allowed to delimit the deficiencies and strengths for timely feedback influencing the process of acquiring skills and the quality of the final product. “

In this research it is proposed to prove that the exercise of education through a virtual classroom with the implementation of a course with b-learning modality and a standardized evaluation system with rubrics favorably affects the quality of the research products and improve the investigative skills of the specialist doctor who is being trained.”

We do not doubt it, since there are investigations, in different areas and fields of knowledge, that have resolved the two statements positively, but of course this investigation has not even seriously proposed it. Notwithstanding the foregoing, it is appreciated that the authors have chosen an interdisciplinary investigation, researching in the field of health, online training and education, and research. In this sense, it is more difficult to present the research in a few pages and cover all the variables, which is why it deserves to be published. Not many examples of interdisciplinary research are presented.

Author Response

Response to Reviewer 3 Comments

Quality of research in residents of medical specialties after a standardized digital training program with rubrics.

Point 1: The article is clear and well written. Easy to read and understand, perhaps due to its linearity. It is observed that authors lacks critical thinking that would have made the article more interesting. Likewise, it can be affirmed that there is interesting information that is not provided and that means that the complexity of the processes it analyzes is not appreciated. For example: The authors explain that “They used the Miller pyramid, hierarchized with four steps that range from cognition to behavior in professional practice, the first step refers to "knowing", the second to "knowing how to apply", the third to "show how it will be done" and the fourth is to "do" as is the case of medical performance in practice. The rubrics are a guide that lists the specific criteria,” We would have liked to know more about the rubrics and how they have applied the steps of the Miller pyramid.

Response 1: The use of Miller's pyramid in medical practice has been expanded and the characteristics and use of rubrics have been deepened. The relationship between the design of the rubrics and the Miller pyramid is explained.

Point 2: A second example is the lack of more information about the students' context that has produced this statement: “The subject of health research has always represented an arid field for the student population, so its development is more related to the motivation of that acquisition” We would like some argument as to why this arid field arises. Don't received students enough capacitation and motivation in the grade?

Response 2: More information about it has been added. In response to your question, students do not receive sufficient training and motivation because they conceive it as something complicated and difficult to do, therefore, when teachers notice this lack of motivation, they try to encourage or stimulate their inclusion in research, however, they receives a negative response in the vast majority of cases.

Point 3: Subsequently, they tell us about five dimensions of the quality research evaluation: the theoretical course, the colloquiums, the final product, the post-seminar evaluation, and the satisfaction survey. And also the design of the Evaluation Rubrics as an Evaluation Instrument composed of a rubric for each topic and specific skill. Presenting some example of the rubrics would have clarified the process, as well as some information on the Post-Seminar Assessment Tool.

Response 3:

The rubric designs and the final satisfaction survey are included as annexes.

Point 4: In the Results section, a table is shown in relation to the medical specialties, when the following table, of the results of the satisfaction survey, is performed globally. They also do not justify why it is done in this way since no argument is derived regarding the results by specialty, we see no reason to present the tables differently.

Response 4: The tables you mention have been removed at the suggestion of other reviewers.

 Point 5: In the presentation of the averages obtained from the population studied, the reason for the low average number of students with publications is not commented: “It is noteworthy that 10.8% of the scholarship holders report having some type of publication before the seminar”.

Response 5:  As explained in the manuscript, there is no relationship between the students' background in research, including participation in previous publications, related to the quality of the work or the dimensions studied.

Point 6: The authors express that “When evaluating the quality of the final project through the rubric, a lower performance was observed in the section referring to "Material and methods". And in the Discussion authors argument “the student ideally requires a clinical tutor accompanied by a methodological tutor, although there is clinical teaching staff, few have the training for the research area.” Given the importance of mastering research methodologies for young researchers at the beginning, and this being a weak point, this fact is not deepened, as it should.

Response 6: This suggested observation has been deepened in detail.

Point 7: Additionally, we find scant mention of the following result, “30 (40.5%) scholarship holders refer to their satisfaction with the development of the course as "bad"". But in the discussion, it is pointed out that “In turn, the satisfaction survey allowed us to visualize the highlights that need to be fed back to improve the research seminar: the format and practicality of the contents, the clarity, and diversity of the activities, the structure of the course, and the interactivity between teachers and students”. No results have been shown in these aspects! The discussion and the conclusions are excessively optimistic: The results of the satisfaction surveys do not contain reasons why the students have expressed that their satisfaction is excellent, good or bad, therefore we do not understand where they have obtained the information to affirm that “However, there are still deficiencies in the program, which thanks to a combination of the perspective of the student and the teacher, allows us to provide feedback and improve our research seminar to provide even higher quality in health education and research”.  We do not doubt the following two statements of the authors:

“Standardized rubrics allowed to delimit the deficiencies and strengths for timely feedback influencing the process of acquiring skills and the quality of the final product. “

In this research it is proposed to prove that the exercise of education through a virtual classroom with the implementation of a course with b-learning modality and a standardized evaluation system with rubrics favorably affects the quality of the research products and improve the investigative skills of the specialist doctor who is being trained.”

Response 7:

It is important to point out that in our study it was observed that four out of ten scholarship holders report poor satisfaction with the development of the course, considering in this dimension the questions were directed to the fulfillment of course objectives, the organization and planning of the course contents, the workload of each topic during the course, the active participation that students present during the course, draws attention to these items. We observed that six out of ten students reported good satisfaction. It is important to highlight that these results allow us to have an approach to how the student perceives the course, however, in order to have an evaluation of the reality of incorporating the standardized item with the rubric of each of the course products that allows objectively evaluate if the student is effectively reaching the competence to generate the learning products, in this way consider as teachers that the satisfaction is sometimes influenced by the student's behavior, the intrinsic motivation and the scope obtained in their own products, which at At the moment of being subjected to an interview or questionnaire, the student exclusively responds considering their own consideration, so it is important that objective measurements are incorporated in all satisfaction studies.

Point 8: We do not doubt it, since there are investigations, in different areas and fields of knowledge, that have resolved the two statements positively, but of course this investigation has not even seriously proposed it. Notwithstanding the foregoing, it is appreciated that the authors have chosen an interdisciplinary investigation, researching in the field of health, online training and education, and research. In this sense, it is more difficult to present the research in a few pages and cover all the variables, which is why it deserves to be published. Not many examples of interdisciplinary research are presented.

Response 8: We appreciate your observation and suggestions. We will improve our work as requested.

Round 2

Reviewer 1 Report

A remarcable effort has been made to improve the manuscript in the suggested direction. Nevertheless it seems that some more effort need to be done. I am convinced that the linguistic barrier is the hardest obstacle that this paper have to pass before being published. I suggest the authors to use a proof reading service. Normally it really helps all those like me and the authors of the manuscript that are a neo-latin language speakers. I made a brief list of changes required and I'd like to revise the paper a third time before accepting it for publication. I put this requests in a separate pdf file.

Author Response

No.

Comentaries

Response  autor

1

Page 3 “The following variables were established and defined: Research quality: variable composed

of five dimensions:” the Authors talk about variables but then only a single 5-dimension variable is

described. Better something like: “Outcome: the efficacy of the learning method was assessed along

five different dimensions”

The suggested change is made.

2

Page 4 line 6 “This survey was designed and validated by pairs using the Delphi method with a 3/3

concordance” by pairs or bay peers?

The text is changed to the wording of experts on the subject

3

Page 4 in the middle of the discussion of point 5: “The process of this method is based on the following characteristics: selection and conformation of the panel of experts: that in our study those

who have a master's or doctorate in health education sciences and previous publications were selected.” review the English in general terms: the entire point 5 must be re-written. In this way is difficult to understand what has been done to achieve concordance among rateres after point 5: “As an intervening variable, the research background of the fellow:”. Better “the research background of the fellows was considered as an intervening variable in the study”.

The wording is modified

4

Page 4: “Design and validation of academic content of the module” only the validation is described.

The wording is modified

5

Page 4: close to the end of the page “in pandemic Restriction of publication” a dot is missing after pandemic.

Add module layout description and fix missing point

6

Page 4 page 5 “ Statistical Analysis: We integrated the data into a base of the SPSS system.

"StatisticalStatisti-cal Package for the Social Sciences" (SPPS version 21). Descriptive statistics

applied: measures of central tendency: mean, median, standard deviation, proportions or

percentagespercent-ages, and it was determined with the Chi-square test for qualitative variables,

and as a hypothesis test the student's t statistic was used for bivariate and ANOVA analysis. of a

factor for different groups, considering the grouping variables satisfaction considering a minimum

significance of 0.05. The results are expressed in bar graphs, histograms, and tables.”

This paragraph is hard to understand. Keep it simple. Something like: “collected data were

submitted to statistical analysis using SPSS 21. Descriptive statistics were computed for all

variables and hypotheses test were run comparing groups via t test or one-way ANOVA for

continuous variables and via Chi-square for qualitative variables.”

The wording is modified

7

Bottom of page 6: I suggest the authors to code the experience in research methodology in the following way: 4 fellows with a publication, 3 fellows with formal education in research

methodology, 2 fellows who completed a introductory course in research methodology, 1 fellows

with no methodological background and to run a Spearman-Brown correlation between this

intervening variable and the grades. Particularly interesting would be the correlation with the score

of the final product.

The recommendation is followed and the suggested statistics are applied

8

Page 7, third line: “A satisfaction survey was applied with a consistency of 0.742 of Cronbach's

alpha in which 74 (87.1%) participants.” ?? hard to understand.

The wording is corrected

Reviewer 2 Report

The manuscript has improved in the sense of adding more context and writing checking. However, more efforts still need to be done, including English editing service and double-checking the statistical testing (i.e., small sample size issue). I suggested more changes required in a separate pdf file and I'd like to see another revision.

A statistical concern is the small group sample size when you split into different specialty groups.

Table 1: some of the cells have less than 5 which probably violates the chi-square assumptions, how did you handle that.

Similarly, some of the specialties have a very small sample size (e.g., < 10), so when you conducted ANOVA, a better approach would be to collapse some of the specialties into one group if possible or drop the very small groups.

Author Response

No.

Comentaries

Response  autor

1

A statistical concern is the small group sample size when you split into different specialty groups.

It is the total number of physicians enrolled in the specialties, therefore it is not a sample

2

Table 1: some of the cells have less than 5 which probably violates the chi-square assumptions, how did you handle that.

In table 1, the Fisher test is estimated for cell values less than 5 and is placed at the bottom.

3

Similarly, some of the specialties have a very small sample size (e.g., < 10), so when you conducted ANOVA, a better approach would be to collapse some of the specialties into one group if possible or drop the very small groups.

Numerical variables with a normal distribution are susceptible to traits along with statistical tests such as ANOVA that may not be available for estimation and that include all students enrolled in majors.